# Evaluation of the Dimensional Stability of Black Poplar Wood Modified Thermally in Nitrogen Atmosphere

**DOI:** 10.3390/ma14061491

**Published:** 2021-03-18

**Authors:** Olga Bytner, Agnieszka Laskowska, Michał Drożdżek, Paweł Kozakiewicz, Janusz Zawadzki

**Affiliations:** The Institute of Wood Sciences and Furniture, 159 Nowoursynowska St., 02-776 Warsaw, Poland; olga_bytner@sggw.edu.pl (O.B.); agnieszka_laskowska@sggw.edu.pl (A.L.); pawel_kozakiewicz@sggw.edu.pl (P.K.); janusz_zawadzki@sggw.edu.pl (J.Z.)

**Keywords:** black poplar, dimensional stability, factor influence, nitrogen atmosphere, thermal modification

## Abstract

Black poplar (*Populus nigra* L.) was thermally modified in nitrogen atmosphere. The effects of the modification process on poplar wood were evaluated for temperatures: 160 °C, 190 °C, and 220 °C applied for 2 h; and 160 °C and 190 °C for 6 h. The percentual impact of temperature and time of modification on the properties of modified wood was analysed. The study permitted the identification correlations between the chemical composition and selected physical properties of thermally modified poplar wood. The dimensional stability of poplar wood was improved after thermal modification in nitrogen. The higher the temperature of modification, the lower the equilibrium moisture content (EMC) of black poplar. At the temperature of 220 °C, EMC was two times lower than the EMC of non-modified black poplar. It is also possible to reduce the dimensional changes of wood two-fold (at the modification temperature of 220 °C), both in radial and tangential directions, independently of the acclimatization conditions (from 34% to 98% relative humidity, RH). Similar correlations have been found for wood that has been soaked in water. Higher modification temperatures and longer processing times contributed to a lower swelling anisotropy (SA).

## 1. Introduction

Thermal modification consists in exposing wood to high temperatures, which changes the chemical composition of wood irreversibly and impacts its physical and mechanical properties [1,2,3,4,5]. Various kinds of thermal modification are applied nowadays in Europe and around the world, among others ThermoWood^®^ from Finland, Retification^®^ and Torrefaction^®^ from France, PLATO-wood^®^ from the Netherlands, and Oil-Heat Treatment (OHT) from Germany. The main difference between these processes consists of the environment in which they are performed, as well as the time of modification. A detailed comparison of wood modification methods, including thermal modification in nitrogen, can be found in selected publications [2,6,7].

One of the methods that is currently being used in the market is modification in a nitrogen atmosphere. This process, initially developed in France by Company NOW (New Option Wood), is equivalent to a mild pyrolysis in inert atmosphere [8]. The process makes use of wood with 12% moisture content, which is slowly heated until reaching the temperature of 200–240 °C in nitrogen atmosphere, where the content of oxygen does not exceed 2%. The process was industrialized in 1997, and the wood is being sold under the Retified Wood trademark. Both deciduous and coniferous wood species can undergo thermal modification [9]. High-density species are more difficult to process with this method than low-density species. With species of high density (mostly hardwood) heat treatment has a tendency to induce cracking, drastically lowering the mechanical properties [8]. Thermal modification of low-density species provides a higher quality surface after machining [10]. The wood species that are usually modified in nitrogen atmosphere are, among others, pine, spruce, birch, poplar, oak, ash, and black locust [11,12]. The main goal of modification is to acquire wood with better dimensional stability and higher durability that can be used in variable weather conditions. Thermally modified wood can be used for furniture veneers [11], floor materials [11,13], and, thanks to a higher resistance to fungi, in external building elements such as facade battens and terrace boards [14], structural elements [4] and to produce materials dedicated for humid environments, etc. [15].

The wood of black poplar (*Populus nigra* L.) has high potential within the scope of biomass production. Black poplar has a large distribution area throughout Europe and is also found in North Africa and Central and West Asia. The distribution area extends from the Mediterranean in the South to around 64° latitude in the North and from the British Isles in the West to Kazakhstan and China in the East. The distribution area also includes the Caucasus and large parts of the Middle East [16,17]. Black poplar is a fast-growing wood species, both in forest and plantation habitats. Moreover, poplar is a species planted in industrially degraded areas in order to revitalize them. Poplar wood is characterized by its low density (below 500 kg × m^−3^) resulting in low resistance parameters and low natural durability. Due to its high porosity and the lack of a clearly formed heartwood, the wood of black poplar has high hygroscopicity and significant dimensional variance in an environment with variable relative air humidity. These characteristics significantly limit the possible applications of black poplar wood [18]. Nowadays, it is used mainly as a resource for the cellulose industry and as a fuel material. Thermal modification in nitrogen atmosphere can significantly increase the functional value of black poplar wood, and as a result change its position in the market for wood and engineered wood materials.

Nguyen et al. [19] conducted research on the physical properties of bamboo after thermal modification in nitrogen atmosphere. The research showed that the equilibrium moisture content (EMC) of modified bamboo was lower compared to unmodified. These changes progressed with the modification temperature. The reduction of EMC was small at 130 °C (0.54% to 0.76%). Larger changes occurred for higher modification temperatures, i.e., 180 °C (3.60% to 4.44%) and 220 °C (5.6% to 5.7%). Heat treatment in nitrogen atmosphere influences EMC, which translates into smaller dimensional changes. Similar dependencies take place in an air atmosphere. With the increase of modification temperature, the dimensional stability of the species modified in air also improved. Studies of white oak (*Quercus alba* L.) showed that a significant improvement of the dimensional stability of the wood was due to the reduction of hydrophilic hydroxyl (-OH) groups [20]. Kubovsky et al. [21] confirmed that chemical changes in the main structural compounds of modified wood (Thermowood) improve the dimensional stability. Esteves et al. [22] found that dimensional stability in the radial and tangential directions (measured as the Anti Shrinking Efficiency, ASE) of eucalyptus wood increased, with maximum values of 88% and 96%, respectively, at 200 °C for air modification.

According to Vernois [8] and Yang et al. [23], operating conditions are essential, and such parameters as atmosphere, temperature, processing time, rate of heating, cooling down duration, species, weight and dimensions of the pieces, as well as moisture content of the wood, strongly affect the final properties of modified wood. Therefore, the selection of heat treatment parameters according to the material of interest is important in producing thermally modified materials, to reach an optimum balance between the improvement in moisture resistance and the decrease of mechanical characteristics for the intended application. Further research in this area is also justified by the fact that there are no models that would provide a precise image of the changes that wood undergoes in variable conditions of the modification process. The main purpose of the study was to determine the relations between the conditions of the thermal process in nitrogen atmosphere and the chemical and physical properties of black poplar (*Populus nigra* L.) wood. Reference literature in this field lacks data on the properties (related in particular to the wood’s dimensional stability) of black poplar wood thermally treated in nitrogen atmosphere, which can improve its dimensional stability.

## 2. Materials and Methods

### 2.1. The Origin and Preparation of Black Poplar Wood

Black poplar (*Populus nigra* L.) wood was used for the study. The black poplar was obtained from a forest in Poland, in the eastern part of the Mazovian province, State Forest District Sokołów Podlaski. It was solid wood from 40-year-old poplars. The trees had a diameter at breast height (DBH) up to 0.5 m and a mean growth ring width greater than 5 mm. Wood without defects such as knots, tangled fibers, cracks, insect trails or rot was used for the tests. The dimensions of the samples used for modification were as follows: 300 mm × 20 mm × 20 mm (L × T × R). The surface of the wood samples was finished by planing. One set of samples was not modified and was treated as a control group. The control samples, as well as samples after different variants of modification, were cut into smaller samples for the purpose of testing individual properties.

### 2.2. Thermal Modification in Nitrogen Atmosphere

The process of thermal modification was carried out with black poplar wood in nitrogen atmosphere. The modification was performed in an 0.25 m^3^ chamber (Explo Solutions sp. z o. o., Warsaw, Poland). The modification chamber is adjusted to work in pressures between -1 atm and 5 atm, is made of acid resistant steel, and is equipped with forced air circulation. The device is controlled by a computer, with the option to carry out a technological process composed of eight steps and lasting up to 168 h.

The modification process took place at temperatures of 160 °C and 190 °C for 2 h and 6 h; and at 220 °C for 2 h. Each kind of modification was performed on 30 samples. The individual modification variants were carried out with a specific time-temperature program controlled by a computer. The first stage of modification consisted in drying the wood for 10 h at a temperature of 110 °C. The next stage consisted in slowly heating the wood (10 °C/1 h) until reaching a temperature of 130 °C, and drying it for 2 h. The subsequent stage consisted in reaching the target temperature (10 °C/1 h) and later the thermal processing of the material took place, by continuing to heat the wood at the constant target temperature for the specified time. The last stage consisted in cooling the material by switching off the heating. The thermal modification programs applied in the study have been presented in Figure 1.

### 2.3. Density, Moisture Content and Mass Loss

The wood density (ρ) was determined according to the ISO 13061-2:2014 [24] standard. Wood moisture content (MC) was measured according to ISO 13061-1:2014 [25]. The mass loss (ML) was determined by comparing the mass of the samples before and after the thermal process in nitrogen atmosphere. The ML was expressed as a percentage of the initial mass of the totally dry wood and was calculated according to Equation (1), where $m_{o}$ is the mass of the oven-dried wood (g), and $m_{m}$ is the mass of the oven-dried wood after thermal modification (g):(1)$$ML=\frac{m_{o}-m_{m}}{m_{o}}\times 100\;\left(\%\right),$$

### 2.4. Chemical Composition

The black poplar wood used for chemical tests had been ground in a laboratory mill, and later the material was sieved with the use of a set of sieves. Chemical tests were performed with a 0.43–1.02 mm fraction of the material. The production of extractives substances was performed with a mixture of chloroform and 96% ethyl alcohol in the ratio 93:7 *w/w* in a Soxhlet apparatus (Glassco Laboratory Equipments, Manglai, India), for 10 h [26]. About 5 g of wood were used for each extraction. Lignin, cellulose and holocellulose content was determined with the use of previously extracted and dried materials in the amount of ca. 1 g. Each measurement was repeated three times. Mass measurements were carried out to a precision of 0.001 g. Cellulose content was determined with the Kürschner-Hoffer method [27]. Holocellulose tests were performed with the use of sodium chlorite and acetic acid, according to the method described by Wise et al. [28]. The content of hemicelluloses was calculated on the basis of the difference between the content of holocellulose and cellulose. The percentual lignin content was calculated with the TAPPI T 222 om-15:2015 method [29]. Moreover, the procedure NREL/TP-510-42618 [30] was used to calculate the content of soluble lignin in the filtrate, using a UVmini 1240 spectrophotometer (Shimadzu, Kyoto, Japan) with a wavelength of 205 nm. The tests were conducted using analytical grades of sulfuric acid solution 95% pure p.a., nitric acid 65% pure, and chloroform. The chemicals were obtained from Chempur (Piekary Śląskie, Poland). The sodium chlorite used was of reagent grade and was purchased from Sigma-Aldrich (Poznań, Poland). The ethanol was of technical grade from Linegal Chemicals (Warsaw, Poland).

### 2.5. Determination of the Equilibrium Moisture Content and Dimensional Changes of Wood

Black poplar wood samples with dimensions of 30 mm × 20 mm × 20 mm (L × T × R) (longitudinal) were used for determining the equilibrium moisture content (EMC). The wood samples, after being dried to a MC of 0%, were placed into containers in which the relative humidity (RH) was 34%, 65%, and 98% at 20 °C (±2 °C). The wood EMC and dimensional changes were measured when the mass of the wood samples remained unchanged over three weighings at 48 h intervals. Wood conditioning (acclimatization) at various RH conditions was achieved using saturated solutions of chemicals, disclosed in Table 1.

The chemicals used (Table 1) were of professional analysis (p.a.) grade and were obtained from Chempur (Piekary Śląskie, Poland). Measurement of the EMC and dimensional changes of black poplar wood were completed when the mass of the wood samples remained unchanged over three weighings at 48 h intervals. The RH was measured using an AZ 9871 anemometer (AZ Instrument Corp., Taichung City, Taiwan). Altogether, one hundred and twenty samples were used for testing sorption (adsorption) properties.

Dimensional changes in the radial ($DCH_{R}$), tangential ($DCH_{T}$) and longitudinal ($DCH_{L}$) directions were determined according to Equations (2)–(4), respectively:(2)$$DCH_{R}=\frac{R_{c}-R_{o}}{R_{o}}\;\times 100\;\left(\%\right),$$
(3)$$DCH_{T}=\frac{T_{c}-T_{o}}{T_{o}}\;\times 100\;\left(\%\right),$$
(4)$$DCH_{L}=\frac{L_{c}-L_{o}}{L_{o}}\;\times 100\;\left(\%\right),$$
where $R_{o}$, $T_{o}$, $L_{o}$ are the dimensions (mm) of the wood samples in oven-dry condition measured in the radial, tangential and longitudinal directions, respectively, and $R_{c}$, $T_{c}$, $L_{c}$ are the dimensions (mm) of the wood samples conditioned at different RH, measured in the radial, tangential, and longitudinal directions, respectively. Sample dimensions were determined with an accuracy of ±0.01 mm.

The changes in wood volume during humidification were calculated from Equation number (5):(5)$$VCH=\frac{VCH_{c}-VCH_{o}}{VCH_{o}}\;\times 100\;\left(\%\right),$$
where $VCH_{o}$ is the volume (mm^3^) of the wood samples at oven-dry condition, and $VCH_{c}$ is the volume (mm^3^) of the wood samples conditioned at different RH.

### 2.6. Determination of Water Absorption and Swelling

Wood samples with dimensions of 20 mm × 20 mm × 20 mm were dried at a temperature of 103 °C ± 2 °C until reaching stable mass, defined as a mass change between two separate measurements not exceeding 0.2%. The tested material was cooled to room temperature (20 °C ± 2 °C) in a desiccator. Subsequently, samples were soaked in distilled water at 20 °C ± 2 °C. The samples were weighed after 5 min, 10 min, 20 min, 30 min, 40 min, 50 min, 60 min, 70 min, 80 min, 90 min, 110 min, 130 min, 3 h, 5 h, 7 h, and 24 h of soaking in water. Moreover, the absorption was determined until no further change in mass occurred (at saturation point) i.e., after reaching maximum moisture content (MMC).

Water absorption (WA) was calculated in accordance with formula 6, where $m_{o}$ is the mass of wood in absolute dry state (g), and $m_{s\left(t\right)}$ is the mass after soaking in water (g) after a specific time (1 h or 24 h, at the saturation point):(6)$$WA\left(t\right)=\frac{m_{s\left(t\right)}-m_{o}}{m_{o}}\;\times 100\;\left(\%\right),$$

At the same time, the same samples were also tested to determine the swelling of black poplar. Sample dimensions were measured in the radial, tangential and longitudinal directions, after 1 h and 24 h of soaking and at the saturation point. Linear swelling in the radial ($S_{R}$) and tangential ($S_{T}$) directions were calculated from formulas (7) and (8); and for the longitudinal direction ($S_{L})$ according to formula (9):(7)$$S_{R}=\frac{r_{c}\;-\;r_{o}}{r_{o}}\;\times 100\%,$$
(8)$$S_{T}=\frac{t_{c}\;-\;t_{o}}{t_{o}}\;\times 100\%,$$
(9)$$S_{L}=\frac{l_{c}\;-\;l_{o}}{l}\;\times 100\%,$$
where $r_{o}$, $t_{o}$, and $l_{o}$ are the dimensions (mm) of wood samples in absolute dry state, measured, respectively, in the radial, tangential and longitudinal direction; and $r_{c}$, $t_{c}$, and $l_{c}$ are the dimensions (mm) of wood samples measured after soaking in water.

The volumetric swelling (*VS*) of black poplar wood after 1 h, 24 h and at saturation point was determined according to equation (10), as:(10)$$VS=\frac{V_{s}-V_{o}}{V_{o}}\;\times 100\;\left(\%\right),$$
where $V_{o}$ is the volume (mm^3^) of the wood samples in oven-dry condition, and $V_{s}$ is the volume (mm^3^) of the soaked wood samples. This calculation was performed with methodology adjusted from appropriate standards: ISO/DIS 13061-13:2016 [31] and ISO/DIS 13061-14:2016 [32].

### 2.7. Statistical Analysis

Statistical analyses were performed using the STATISTICA version-12 software (TIBCO Software Inc., Palo Alto, CA, USA). The statistical analysis of the results was based on the *t*-test or the ANOVA (Fischer’s F-test), with a significance level *(p)* of 0.050. On the basis of the sum of squares (SS), we calculated the percentual impact of the analysed factors (temperature and time of modification), the so called Factor Influence on the chemical and physical properties of black poplar wood thermally modified in nitrogen atmosphere.

## 3. Results and Discussion

### 3.1. Changes in Chemical Composition

As a result of thermal modification in nitrogen atmosphere, the content of structural and non-structural compounds in black poplar wood changed. The most significant changes were observed in the case of the content of hemicelluloses and chloroform-ethanol extractives, and were less significant for cellulose and holocellulose. On the other hand, no statistically significant changes have been observed for the lignin content in the wood before and after the modification process (Table 2). Similar results concerning the changes in the content of individual components were also observed for pedunculate oak (*Quercus robur* L.) modified in air atmosphere [33].

Depending on the temperature and time of modification, the content of hemicelluloses oscillated between ca. 3% and ca. 26%, and was lower than in non-modified black poplar wood. Hemicelluloses, as polysaccharides with low degree of polymerization and amorphic structure, undergo degradation in temperatures higher than 140–150 °C [34], which translates into the correlation of their loss at higher modification temperatures. This has also been confirmed by statistical data (Table 3), which suggest that modification temperature was 95% responsible for the changes in hemicelluloses content, while the time of modification was insignificant (an impact at the level of 1%). The highest reduction of hemicellulose content was observed for black poplar wood modified at a temperature of 220 °C and for a time of 2 h. The content of hemicelluloses in modified black poplar wood in these conditions was 10 times lower than in non-modified wood. Additional elements accelerating the degradation of hemicelluloses are the acetyl groups and carboxyl groups they contain. The temperature increase causes those groups to detach and create formic acid and acetic acid. These acids further accelerate the degradation of wooden materials [35,36,37]. The degradation of hemicelluloses to simpler organic compounds causes an increase in the share of chloroform-ethanol extractives. The higher the modification temperature and the longer the time of modification, the higher the concentration of chloroform-ethanol extractives in black poplar wood, and the differences were statistically significant (*t*-test, *p* ≤ 0.050). The only exception was modification at 160 °C for 2 h, after which the content of extractive compounds in black poplar wood grew by ca. 13% compared to non-modified wood, and the differences were not statistically significant (*t*-test, *p* > 0.050). Only an increase of extractive compounds content by 26% in black poplar wood after modification at 160 °C for 6 h resulted in statistically significant changes of the compound under analysis. In the case of black poplar wood modified at a temperature of 220 °C and a time of 2 h, we observed a 3.5 times higher content of chloroform-ethanol extractives than in the case of non-modified wood. It is worth noting that 74% of the changes in chloroform-ethanol extractives depended on modification temperature, while time of modification was responsible for 15% of these changes (Table 3).

Cellulose, as a polysaccharide with a higher degree of polymerization and more complex amorphic-crystalline structure, is more resistant to high temperatures, and as a result its degradation happens much more slowly than in case of hemicelluloses. The increase in the cellulose content was relative and resulted from the degradation of hemicelluloses, which translated into a higher share of this sugar in the wood, whose mass was reduced. This relative increase, depending on the temperature and time of modification, amounted to 5–18% (Table 2).

Among the compounds under analysis, the most thermally stable compound in the wood of black poplar was lignin, and the analysed correlations are in line with the literature data [38,39]. The increase of lignin content in modified wood was relative and the differences were not statistically significant (Table 2). Compared to hemicelluloses and cellulose, lignin has better hydrophobicity and chemical reaction inertness because of its lower hydroxyl content. As the most heat-resistant component, the slight increase in lignin content may be due to the condensation and cross-link reactions of lignin or the production of compounds featuring aromatic ring products induced by heat treatment [40]. Moreover, a large amount of acetic acid produced by the thermal degradation of hemicelluloses can possibly catalyze the degradation of cellulose in the amorphous region. Lignin is esterified under the catalysis of acidic reagents. As a result, the number of hydroxyl groups decreases and the number of carbonyls increases, which is equivalent to replacing the hydroxyl group by the carbonyl group with weak hygroscopicity [41,42].

### 3.2. Mass Loss and Density of Black Poplar Thermally Modified in Nitrogen Atmosphere

In general, it can be concluded that the higher the modification temperature, the greater the mass loss (ML) of black poplar wood (Table 4). However, statistically significant differences (compared to non-modified black poplar) were observed for wood modified at 190 °C for 6 h and 220 °C for 2 h. The ML for the analysed modification variants amounted to ca. 2% and 7%. These changes were a result of the degradation of wood components, mainly the hemicelluloses, caused by high temperatures (Table 2). Literature data suggest that ML is basically the effect of evaporation of non-structural wood components, such as terpenes, fat, wax and phenols, and the decomposition of the least stable hemicelluloses [43,44]. At the same time, it should be noted that the degree of wood degradation, and especially changes in the content of chloroform-ethanol extractives, also depend on the time of thermal treatment, which is confirmed by reference literature data [45]. Bal [46] proved that ML is more significant at higher modification temperatures. The ML after thermal modification of pine wood in nitrogen atmosphere amounted to 0.8% for 180 °C, 1.3% for 200 °C, and 2.9% for 220 °C.

On the basis of former research [37], it should be mentioned that the modification of black poplar wood in superheated steam was more destructive to the wood’s mass than thermal modification in nitrogen atmosphere. The ML of black poplar wood modified in superheated steam during 2 h at temperatures of 160 °C, 190 °C, and 220 °C amounted to, respectively, 3%, 4%, and 12%. On the other hand, the ML of black poplar wood modified in nitrogen atmosphere with the same time and temperature parameters amounted to, respectively: 1%, 1%, and 7%.

The density of non-modified black poplar wood in an absolute dry state amounted to 375 (±38) kg x m^−3^ (Table 4). This is the typical density of poplar wood cut at a young age (around 40 years) characterized by a high share of juvenile wood. As a result of thermal modification in nitrogen, there was a mass loss that translated into a lower density. It has to be noted that a significant reduction of density (about 7%) was observed for black poplar modified at the temperature of 220 °C for 2 h (*t*-test, *p* ≤ 0.050). The indicated relations are confirmed by literature data. Bal [47] achieved statistically insignificant changes in poplar wood density (*Populus x euramericana I-214*) at lower temperatures, and only the highest temperature (200 °C) significantly affected the density of wood after thermal modification. Esteves et al. [2] state that the density of spruce and beech wood after thermal modification at temperatures between 200 °C and 260 °C dropped by 15% and 1%, respectively.

### 3.3. Equilibrium Moisture Content and Dimensional Changes of Black Poplar during Humidification at Different Relative Humidities

The equilibrium moisture content (EMC) of black poplar wood achieved in different acclimatization conditions has been presented in Table 4. The higher the modification temperature in nitrogen atmosphere, the lower the EMC of black poplar, and the differences were statistically significant (*t*-test, *p* ≤ 0.050). A longer thermal modification of black poplar wood in nitrogen atmosphere did not significantly affect the changes in EMC. Similar results were published by Dong et al. [48] in a study of the EMC of fast-growing poplar wood after thermal processing, Kozakiewicz et al. [37] for black poplar after thermal modification in superheated steam, and Brito et al. [49] for yellow poplar modified in atmospheric air. In case of black poplar wood modified at 220 °C, the EMC was about two times lower than the EMC of non-modified black poplar. These results are in line with previous studies. Kamdem et al. [39] thermally modified beech wood in nitrogen atmosphere at temperatures of 200 °C and 260 °C. The following EMC reductions were observed: from 10% to 5%, from 14.5% to 8%, and from 21.8% to 12%, at relative air humidity of 66%, 86% and 100%, correspondingly. The changes in EMC result from the degradation of hemicelluloses and areas of amorphic cellulose, caused by the high temperature of the treatment process [2]. This phenomenon has also been observed in this study (Table 2).

Depending on the acclimatization conditions, modified black poplar wood underwent dimensional changes (Table 5 and Table 6). The reduction of dimensional changes was more significant for higher temperatures during the thermal modification process. As a result of thermal modification in nitrogen, it is possible to reduce the dimensional changes of black poplar even by two-fold (at the modification temperature of 220 °C), both in radial direction (DCH_R_), and tangential direction (DCH_T_). The reduction of dimensional changes in the longitudinal direction (DCH_L_) was not significant, even for black poplar modified at 220 °C, and amounted to ca. 0.1% compared to non-modified wood (Table 6).

A larger impact of the processing was visible in the case of changes in the volume (VCH) of black poplar wood during the humidification process (Table 6). In case of black poplar modified at a temperature of 220 °C, we observed VCH values that were at least two times lower for each climate under research than in the case of non-modified poplar. Chemical processes explain the reduction of dimensions after thermal modification. The exposure of wood to high temperatures causes a significant degradation of hemicelluloses. The research of Liang and Wang [50] shows that the hydrophobicity of hemicelluloses after nitrogen modification is higher. Additionally, Burmester [51] describes this exact phenomenon as crucial for improving dimensional stability.

When considering the percentage impact of technological factors, it should be noted that modification temperature has a significant influence on the density (ρ), equilibrium moisture content (EMC) and dimensional changes (DCH) of black poplar thermally modified in nitrogen atmosphere (Figure 2). This influence, depending on the value under analysis, fell in a range between 6–97%. The lowest percentual value (6%) of temperature influence was observed for density, while the remaining properties were impacted from 8 to 16 times more. The modification temperature is responsible for about 90% of changes in EMC, independently of the relative air humidity (RH) level. It is important that the modification temperature was more responsible for the dimensional changes of black poplar wood in the tangential direction (DCH_T_, influence at the level of 73–85%) than in the radial direction (DCH_R_, influence at a level of 46–71%), and this influence was more pronounced for black poplar wood humidified in a climate with higher RH. A significant influence of the modification temperature on the dimensional changes in a longitudinal direction (DCH_L_) was observed for black poplar wood acclimatized in 65% RH and 98% RH (impact at a level of 20% and 32%, respectively). On the other hand, acclimatization in 34% RH did not cause significant changes in the longitudinal direction, independently of the modification temperature (Table 6). Considering the impact of modification temperature on the volumetric changes (VCH) of black poplar wood during the humidification process, it can be concluded, in general, that the impact was at a level of 70%-90%, and was a result of the influence, in particular, on the dimensional changes in the tangential direction. It should be noted that the time of modification at the tested levels of variability (2 h and 6 h) did not significantly influence the properties of black poplar, which was especially visible in the low percentual values falling in the range between 0–9% (Figure 2).

### 3.4. Dimensional Changes of Black Poplar after Soaking in Water

Figure 3 and Figure 4 present the moisture content (MC) values of wood after soaking in water. At the beginning, the poplar wood samples were in an absolute dry state, before they were submerged in distilled water, and later their water content was determined after a specific time. Thermal modification of poplar wood (in so called mild conditions, i.e., temperature of 160 °C) did not affect the water-soaking dynamics and the final, maximum moisture content achieved in this process (MMC of poplar wood: 239% for modification time of 2 h and 243% for modification time of 6 h). As a result of temperature increase in the wood modification process, the soaking process slowed down significantly (Figure 3). However, finally (apart from the two last variants: thermal modification at 190 °C for 6 h, and 220 °C for 2 h), the thermal modification process did not impact the MMC achieved (values at a level of 243%, Figure 4).

In general, it can be concluded that, depending on the modification parameters, MMC was from 4 to 7 times higher than WA after 1 h and from 3 to 4 times higher than WA after 24 h of soaking the black poplar wood samples in water (Figure 4). The highest multiplier values were achieved for black poplar modified at the temperature of 220 °C over 2 h. Slower water absorption after a stronger modification in nitrogen atmosphere (temperatures above 190 °C) will have a practical effect in technological processes such as gluing or surface finishing. This is suggested by, among others, results concerning the thermal modification of common beech [52] and black poplar wood in an atmosphere of superheated steam [37].

Figure 5a,b present the swelling values achieved for black poplar and measured in the radial direction (S_R_) and tangential direction (S_T_). It should be noted that the process of thermal modification of wood in nitrogen atmosphere at 160 °C for both time variants did not affect the results or the dynamics of the swelling process. The values of S_R_ and S_T_ were, in these two cases, similar to the values observed for non-modified black poplar wood, and as such, they corresponded to the ranges published in the literature [18]. Higher modification temperatures (190 °C and 220 °C instead of 160 °C), and longer times for the modification process (increasing from 2 h to 6 h) caused lower values of linear swelling (Figure 5a,b), and, as a result, also of volumetric swelling (Figure 6). The strongest stability of wood was achieved after thermal modification at 220 °C. In this case, the S_R_ and S_T_ values were ca. two times lower in comparison with non-modified black poplar wood.

It is important that, as a result of thermal modification in nitrogen atmosphere, we observed a lower swelling anisotropy. The higher the temperature of modification and the longer the process time, the smaller the ratio between tangential swelling and radial swelling, i.e., S_T_*/*S_R_. In the case of non-modified black poplar wood after the first hour of soaking, S_T_*/*S_R_ amounted to 2.7, while with MMC, it amounted to 3.0. On the other hand, the S_T_*/*S_R_ for wood modified at a temperature of 220 °C after the first hour of soaking, amounted to 1.8, and, with MMC, 2.5. In case of poplar wood (*Populus x euramericana I-214*) thermally modified in air at atmospheric pressure, the most significant changes were observed for the process of modification at 200 °C [47].

The values of volumetric swelling (VS) of black poplar wood after different variants of thermal modification have been presented in Figure 6. Roughly speaking, VS can be described as the sum of radial, tangential and longitudinal directions (the latter is usually omitted in engineering practice, because the changes caused by moisture content in this direction are negligible). Both in the case of non-modified black poplar wood, as well as of wood that underwent modification, the VS at MMC was two times higher than after 1 h of soaking in water and ca. 1.3 times higher than the VS after 24 h of soaking. When analysing the values of complete swelling, black poplar wood should be classified as moderately swelling wood and, after thermal modification at 220 °C, as wood with low swelling, which can be interpreted as a beneficial change.

Thermal modification in a nitrogen atmosphere definitely improves the dimensional stability of black poplar wood, similarly to its thermal modification in superheated steam [37], but only in more intense variants of this process that take place at higher temperatures, such as 190 °C and 220 °C. An improved dimensional stability was also achieved in the case of the fast-growing wood species *Acacia mangium*, thermally modified in an air environment [53]. The results obtained are in line with the literature data: more intense processing and longer times of modification increase the dimensional stability of wood; and when comparing the impact of time and temperature, temperature plays a more important role [54].

When we know the density of wood in an absolute dry state and the VS value, we can estimate the fibre saturation point (FSP) according to the formula: FSP ≅ VS/(wood density/water density). The FSP values estimated in this manner are significantly higher than the measured ones (for 98% of RH), but they do have an analogous tendency to the one observed to reduce FSP, together with higher temperatures of thermal modification, which results from a faster percentual reduction of VS in comparison with wood density.

Percentual influence of technological factors, i.e., temperature and time of modification, on water absorption (WA), linear swelling (S), and volumetric swelling (VS) of thermally modified black poplar has been presented in Figure 7. 

The modification temperature shows a significant influence on the WA of poplar wood, depending on the time after which this property was measured. After 1 h and 24 h of soaking in water, this influence was at the level of ca. 75%, and at MMC it was two times lower. The temperature of modification has an important impact on the S_R_ and S_T_ (impact at the level of 50–80%), while a higher influence was observed in the case of S_T_. Along the radial section, layers of earlywood and latewood are placed alternately between each other. Earlywood is characterised by a lower density than latewood. This alternating placement attenuates the swelling. The highest swelling values are observed in the tangential direction, which is caused by the “spherical” orientation of latewood layers, which undergo important dimensional changes. This happens due to a tangential elongation of the cellular lumens. Additionally, the middle lamella has a significant influence on the value of swelling, while we should bear in mind that its total thickness is larger in the tangential direction than in the radial direction, and that it is made, in large part, of hemicelluloses, that undergo degradation very easily during the thermal modification of wood [55,56]. No significant impact was observed between the temperature of modification and S_L_ values. A change of cellulose particle dimensions in the longitudinal direction is hindered by strong glycosidic bonds in cellulose chains, especially when the particles are placed in parallel to the longitudinal axis of the fibre. As a result, the dimensional changes along the fibres (parallel to the grain) are small. Due to a very small swelling in this anatomical direction, it is not taken into account. In ca. 70% of cases, the modification temperature was responsible for the changes of VS of black poplar wood. The modification time did not exert a significant influence on any of the analysed wood properties related to water absorption.

## 4. Conclusions

On the basis of the performed tests, it was proven that thermal modification in a nitrogen atmosphere has an important impact on the chemical composition and physical properties of black poplar wood. The modification temperature has an important influence on the content of structural compounds (except for lignin), chloroform-ethanol extractives and all the physical properties of wood that were analysed (apart from dimensional changes in a longitudinal direction during humidification and soaking in water). Furthermore, the modification time only had a significant influence on the content of chloroform-ethanol extractives. As a result of thermal modification in nitrogen, a loss in mass occurred, which translated into a lower density of black poplar wood. However, a significant density reduction by ca. 7% was observed for black poplar wood modified at a temperature of 220 °C over 2 h. As a result of the modification process, the wood had lower EMC values, lesser dimensional changes, and less swelling anisotropy. The higher the modification temperature, the lower the ratio between static swelling and radial swelling. These correlations resulted, most of all, from a lower content of hemicelluloses that, when exposed to high temperatures, underwent the most extensive degradation of all the wood components under analysis. During the first seven hours of soaking in water, the absorption dynamics slowed down in case of wood modified at 190 °C and 220 °C. On the other hand, the MMC was similar to non-modified wood (differences at the level of 7 percentage points). The achieved results can be useful for the industry and can form part of a larger data base. This research can elucidate the use of thermally processed poplar wood in places where resistance to water is important e.g., elevations, fences. The high dimensional stability of wood is also important in changing room climate conditions for applications such as furniture, wall panels or battens.

## Figures and Tables

**Figure 1 materials-14-01491-f001:**
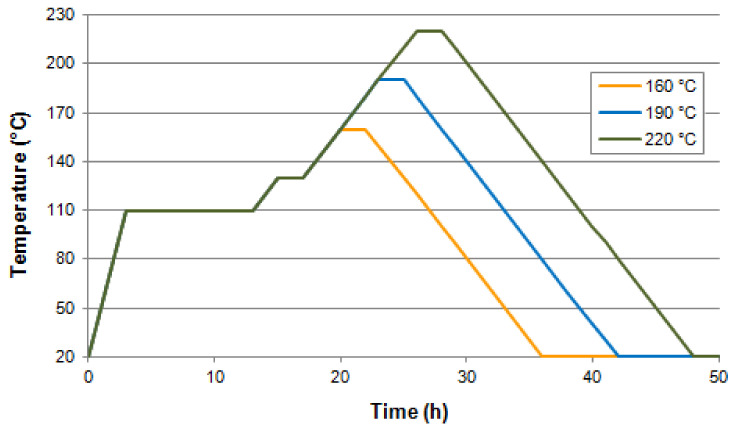
The course of thermal modification in nitrogen atmosphere of black poplar wood.

**Figure 2 materials-14-01491-f002:**
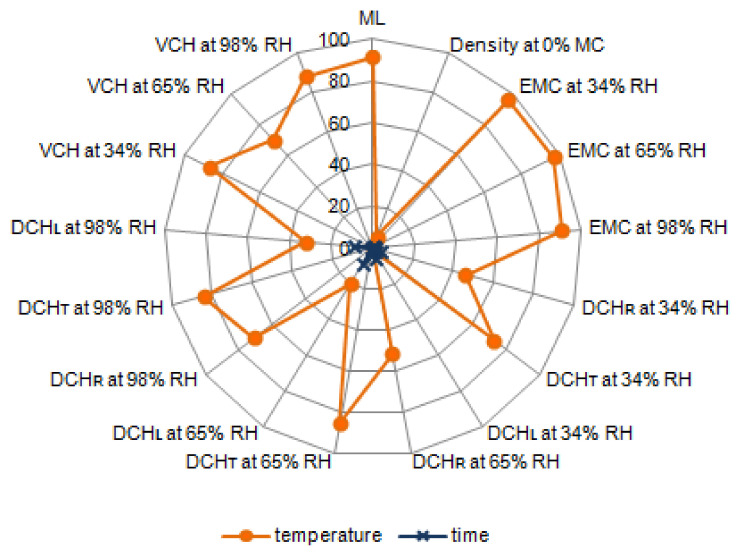
The percentual influence of temperature and modification time on the mass loss (ML), density (ρ), equilibrium moisture content (EMC), dimensional changes in the radial (${\mathrm{DCH}}_{R}$), tangential (${\mathrm{DCH}}_{T}$), longitudinal (${\mathrm{DCH}}_{L}$) directions and volumetric changes (VCH) of black poplar thermally modified in nitrogen atmosphere.

**Figure 3 materials-14-01491-f003:**
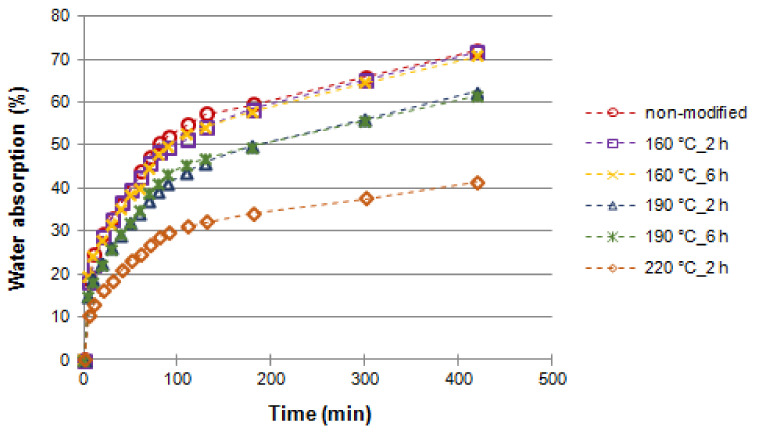
Water absorption (WA) of black poplar thermally modified in nitrogen atmosphere during the first 7 h of soaking.

**Figure 4 materials-14-01491-f004:**
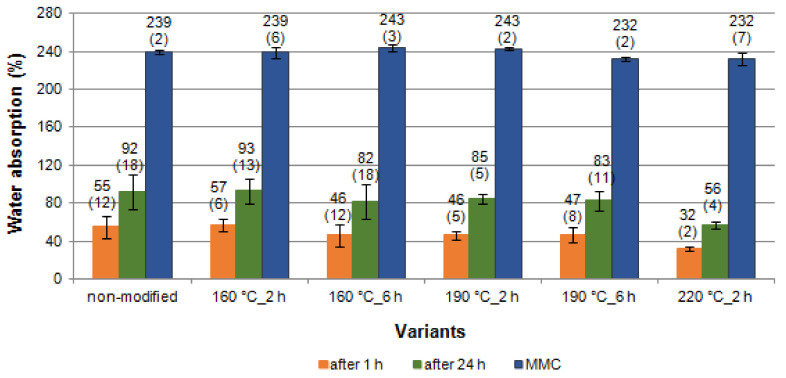
Water absorption (WA) of black poplar thermally modified in nitrogen atmosphere (error bars—standard deviation (±SD)).

**Figure 5 materials-14-01491-f005:**
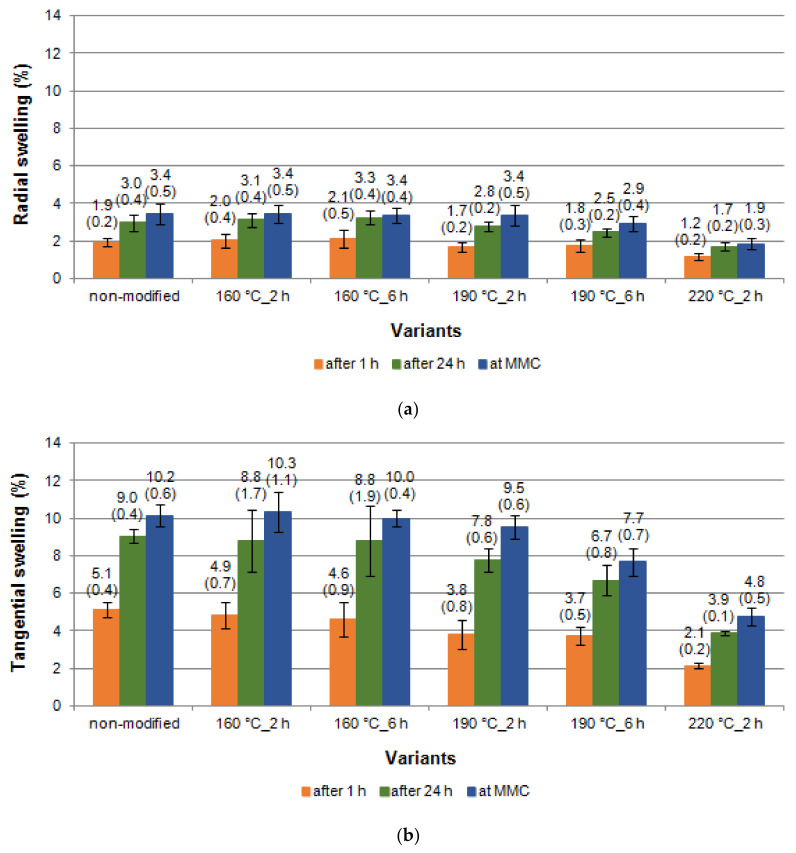
Swelling in the radial (**a**) and tangential (**b**) directions of black poplar thermally modified in nitrogen atmosphere (error bars-—standard deviation (±SD)).

**Figure 6 materials-14-01491-f006:**
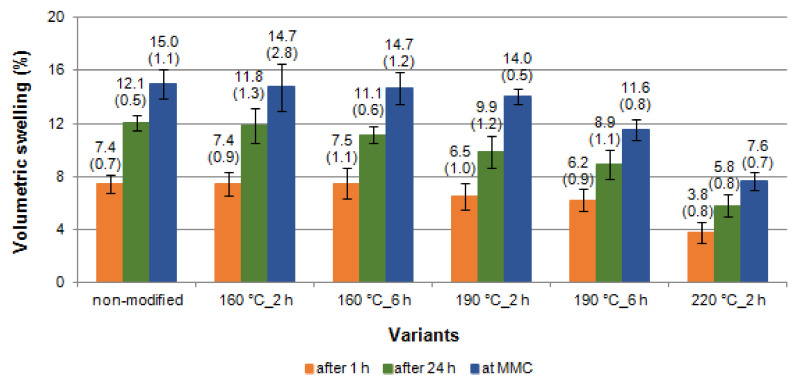
Volumetric swelling (VS) of black poplar thermally modified in nitrogen atmosphere (error bars—standard deviation (±SD)).

**Figure 7 materials-14-01491-f007:**
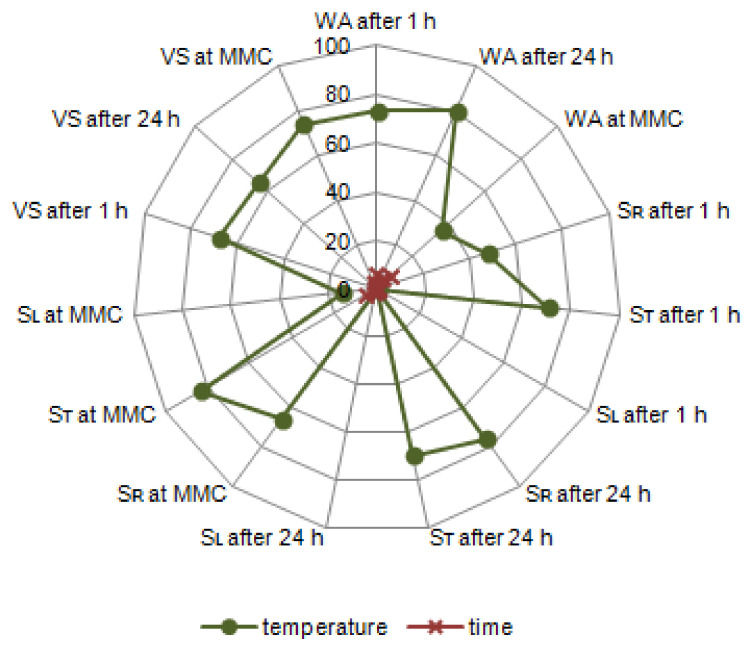
The percentual influence of temperature and modification time on water absorption (WA), linear swelling in the radial ($S_{R}$), tangential ($S_{T}$), longitudinal ($S_{L}$) directions and volumetric swelling (VS) of black poplar thermally modified in a nitrogen atmosphere.

**Table 1 materials-14-01491-t001:** Saturated solutions of chemicals used to obtain appropriate relative humidity (RH).

Saturated Solution of Chemicals		RH (%)
MgCl_2_ × 6H_2_O	Magnesium chloride hexahydrate	34
NaNO_2_	Sodium nitrite	65
K_2_SO_4_	Potassium sulfate anhydrous	98

**Table 2 materials-14-01491-t002:** Chemical composition of black poplar: non-modified and thermally modified in nitrogen atmosphere; ±(SD).

Modification Temperature (°C)	Modification Time (h)	Cellulose (%)	Holocellulose (%)	Hemicelluloses (%)	Lignin (%)	Chloroform-Ethanol Extractives (%)
non-modified	-	52.15 ± 0.49	82.11 ± 0.34	29.96 ± 0.83	24.16 ± 0.48	1.80 ± 0.03
160	2	54.58 ± 0.01 *	78.54 ± 0.87 *	23.96 ± 0.88 *	25.15 ± 0.17	2.04 ± 0.18
	6	54.43 ± 0.68 *	80.78 ± 0.17 *	26.35 ± 0.86 *	24.88 ± 0.87	2.26 ± 0.33 *
190	2	53.82 ± 0.11 *	77.03 ± 0.44 *	23.21 ± 0.55 *	24.85 ± 0.35	2.92 ± 0.13 *
	6	54.81 ± 0.05 *	73.01 ± 0.38 *	18.19 ± 0.43 *	24.81 ± 0.26	6.50 ± 0.15 *
220	2	61.63 ± 0.25 *	64.81 ± 0.31 *	3.18 ± 0.56 *	25.23 ± 0.54	6.59 ± 0.04 *

* Statistically significant differences, based on the *t*-test (*p* ≤ 0.050, control group—non-modified black poplar wood).

**Table 3 materials-14-01491-t003:** ANOVA evaluation of the factors influencing the chemical composition of thermally modified in nitrogen atmosphere black poplar (Fischer’s *F*-test; *p* ≤ 0.050).

Wood Components	Factor	Sum of Squares	Fisher’s *F*-test	Significance Level	Factor Influence (%)
		SS	*F*	*p*	
Cellulose	Intercept	39,045.78	341,029.2	0.000000	-
	Temperature (1)	110.03	480.5	0.000000	98
	Time (2)	1.41	12.4	0.004842	1
	Error	1.26	-	-	1
Holocellulose	Intercept	64,008.36	20,544.73	0.000000	-
	1	404.42	64.90	0.000001	92
	2	3.08	0.99	0.341417	1
	Error	34.27	-	-	7
Hemicelluloses	Intercept	3082.142	810.4047	0.000000	-
	1	891.493	117.2026	0.000000	95
	2	7.987	2.1001	0.175195	1
	Error	41.835	-	-	4
Lignin	Intercept	7501.667	133,683.4	0.000000	-
	1	0.182	1.6	0.241368	21
	2	0.087	1.5	0.239712	10
	Error	0.617	-	-	69
Chloroform -Ethanol Extractives	Intercept	277.3767	366.5698	0.000000	-
	1	53.4715	35.3329	0.000016	74
	2	10.5281	13.9136	0.003323	15
	Error	8.3235	-	-	11

**Table 4 materials-14-01491-t004:** Mass loss (ML), density (ρ), equilibrium moisture content (EMC) at different relative humidity (RH) determined for black poplar, non-modified and thermally modified in nitrogen atmosphere; ±(SD).

Modification Temperature (°C)	Modification Time (h)	ML(%)	ρ at 0% MC (kg × m^−3^)	RH (%)		
				34	65	98
				EMC (%)		
non-modified	-	-	375 ± 38	5.8 ± 0.1	10.0 ± 0.2	27.0 ± 0.9
160	2	0.9 ± 0.5	374 ± 34	5.4 ± 0.1 *	9.1 ± 0.1 *	26.1 ± 1.3
	6	0.7 ± 0.4	374 ± 36	5.0 ± 0.2 *	9.3 ± 0.1 *	25.9 ± 1.4
190	2	0.8 ± 0.4	373 ± 37	4.1 ± 0.1 *	8.3 ± 0.3 *	23.7 ± 1.2 *
	6	2.1 ± 0.7*	366 ± 37	4.1 ± 0.2 *	7.9 ± 0.2 *	22.7 ± 1.0 *
220	2	6.8 ± 0.9*	349 ± 34 *	2.9 ± 0.2 *	5.4 ± 0.1 *	16.0 ± 0.6 *

* Statistically significant differences based on the *t*-test (*p* ≤ 0.050; control group—non-modified black poplar wood).

**Table 5 materials-14-01491-t005:** Dimensional changes in radial (DCH_R_) and tangential (DCH_T_) directions of black poplar, non-modified and thermally modified in nitrogen atmosphere, during humidification at different relative humidities (RH); ±(SD).

Modification Temperature (°C)	Modification Time (h)	RH (%)					
		34		65		98	
		Dimensional Changes (%)					
		DCH_R_	DCH_T_	DCH_R_	DCH_T_	DCH_R_	DCH_T_
non-modified	-	0.9 ± 0.1	1.4 ± 0.2	1.0 ± 0.2	2.8 ± 0.3	2.9 ± 0.2	8.6 ± 0.4
160	2	0.7 ± 0.1 *	1.4 ± 0.1	1.1 ± 0.2	2.5 ± 0.2	2.9 ± 0.4	8.1 ± 0.7
	6	0.6 ± 0.1 *	1.4 ± 0.3	1.1 ± 0.1	2.6 ± 0.3	2.7 ± 0.4	8.2 ± 0.4
190	2	0.6 ± 0.1 *	1.0 ± 0.1 *	1.0 ± 0.1	2.2 ± 0.1 *	2.5 ± 0.3	7.3 ± 0.7 *
	6	0.6 ± 0.1 *	1.0 ± 0.2 *	1.0 ± 0.2	2.3 ± 0.2 *	2.5 ± 0.2	6.5 ± 0.4 *
220	2	0.5 ± 0.1 *	0.6 ± 0.1 *	0.6 ± 0.2 *	1.2 ± 0.2 *	1.4 ± 0.3 *	4.4 ± 0.6 *

* Statistically significant differences based on the *t*-test (*p* ≤ 0.050; control group—non-modified black poplar wood).

**Table 6 materials-14-01491-t006:** Dimensional changes in longitudinal direction (DCH_L_) and volumetric changes (VCH) of black poplar, non-modified and thermally modified in nitrogen atmosphere, during humidification at different relative humidity (RH); ±(SD).

Modification Temperature (°C)	Modification Time (h)	RH (%)					
		34		65		98	
		Dimensional Changes (%)					
		DCH_L_	VCH	DCH_L_	VCH	DCH_L_	VCH
non-modified	-	0.2 ± 0.1	2.4 ± 0.2	0.3 ± 0.1	4.0 ± 0.8	0.4 ± 0.1	11.9 ± 0.5
160	2	0.2 ± 0.1	2.4 ± 0.3	0.3 ± 0.1	3.9 ± 0.4	0.5 ± 0.1	11.6 ± 1.1
	6	0.2 ± 0.1	2.4 ± 0.3	0.3 ± 0.1	3.9 ± 0.6	0.4 ± 0.1	11.6 ± 0.5
190	2	0.2 ± 0.1	1.7 ± 0.2 *	0.2 ± 0.1 *	3.3 ± 0.2	0.4 ± 0.1	10.6 ± 0.8 *
	6	0.2 ± 0.1	1.8 ± 0.3 *	0.2 ± 0.1 *	3.5 ± 0.3	0.4 ± 0.1	10.3 ± 0.7 *
220	2	0.2 ± 0.1	0.8 ± 0.1 *	0.2 ± 0.1 *	2.2 ± 0.4 *	0.3 ± 0.1	6.3 ± 0.5 *

* Statistically significant differences based on the *t*-test (*p* ≤ 0.050; control group—non-modified black poplar wood).

## Data Availability

The data presented in this study are available on request from the corresponding author.

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
