# Peer review of "Evaluation of the Dimensional Stability of Black Poplar Wood Modified Thermally in Nitrogen Atmosphere"

_materials, 2021, doi:10.3390/ma14061491_

Round 1

Reviewer 1 Report

please see my comments in the attached file.

Author Response

Thank you very much for your valuable comments. The article was corrected. Changes were marked in green in manuscript. 

Thank you very much for your valuable comments. The article was corrected. Changes were marked in green.

Best regards

Reviewer 2 Report

„Evaluation of the Dimensional Stability of Black Poplar Wood Modified Thermally in Nitrogen Atmosphere” describes the effect of thermal treatment on wood dimensional stability. However, this effect has already been recognised and is being known for years, and there is plenty of research data available in the literature. It is important then to clearly state what is the novelty of the presented research and include/emphasise this not only at the end of an introduction, as the Authors did, but also in a title, an abstract, and a discussion. It will be helpful for readers to recognise the most important issues and focus on novelty.

I have some more comments and suggestions for the paper:

Abstract:

  • The information that treatment in a nitrogen atmosphere can improve the dimensional stability of black poplar is repeated twice in an abstract (lines 10-11 and 16-17) – please remove one of the sentences.

Introduction:

  • Line 29 – do the Authors mean chemical composition or wood structure (macrostructure, microstructure)? Please, correct and specify.
  • There is plenty of literature related to the effect of thermal modification n wood properties – please provide some more recent references here (lines 28-30).
  • The sentence “Compared to other trees like oak, for example, that take more than 67 100 years, poplars are very fast” (lines 67-68) is hard to understand. Please, modify it to make it clear.
  • I cannot see the relevance of the information given in the third and fourth paragraphs of the Introduction (lines 56-64 and 65-75) to the topic of the paper. Please, remove or re-write these parts to make them clearer. Maybe it would be better to include some of the most important facts from these paragraphs in the next one (lines76-92)?
  • Line 77 – it is not a local wood species – “not local” in respect to which region?
  • Lines 102-104 – which properties? Please, specify.

Materials and Methods:

  • What were the sample dimensions during the treatment? Were the samples cut into smaller specimens before or after the treatment?
  • How many samples were used for the evaluation of chemical composition?
  • How much wood powder was used for the analysis – please, clarify the statement in lines 156-158.
  • Line 154 – what does it mean that extractives were tested with a mixture of chloroform etc.? Please, clarify.
  • Lines 185-186 – 120 samples per each treatment variant or 120 all together?

Results and Discussion:

  • Lines 241-243 – are these results by the Authors as a part of a study or by others? If by others, please add a relevant reference here. If by the Authors, please add the information about testing oak to the paper.
  • Table 2 – do the statistically significant differences relate to the same temperature and different time or the same time but different temperatures, or maybe to untreated wood? Please, specify, because this is not clear.
  • Line 508 – oxygen links – do the Authors mean hydrogen bonds?
  • Since the applied type of wood modification is intended to improve wood performance, please discuss how the observed changes in wood chemical composition and moisture properties can affect its other properties, particularly mechanical behaviour. It seems quite important from the industrial perspective.
  • It would be worth to discuss here if the observed effect of the thermal treatment on the selected properties of black poplar is similar to other wood species or maybe different ? (based on the literature data).

Conclusions:

  • There is no conclusion about the practical issues of the research performed. Please, add some information about the potential application of the results obtained.

Author Response

Thank you very much for your valuable comments. The article was corrected and changes are marked in green. All changes are marked in attachment

Best regards

Reviewer 3 Report

The article analysis the thermal modification method in nitrogen atmosphere of poplar wood. The thematic is of great interest and relevance. The results presented are scientifically sound, the discussion made is interesting and statistical analysis made shows the quality of the paper. The conclusions proposed corresponds to the results of the research. I have almost no comments, except one:

1) The abstract need to be rewritten, because it lack of specification of the problem, methods to solve the problem and the main results obtained during research.

Author Response

(The authors gave the same response as above.)
